# The synthetic future of algal genomes

Hugh D. Goold,[1,2,3] Jeffrey L. Moseley,[4,5,6] and Kyle J. Lauersen[7,*]

[1]New South Wales Department of Primary Industries, Orange, NSW 2800, Australia
[2]ARC Center of Excellence in Synthetic Biology, Macquarie University, Sydney, NSW 2109, Australia
[3]School of Natural Sciences, Macquarie University, Sydney, NSW 2109, Australia
[4]California Institute for Quantitative Biosciences, University of California, Berkeley, Berkeley, CA 94720, USA
[5]Division of Environmental Genomics and Systems Biology, Lawrence Berkeley National Laboratory, Berkeley, CA 94720, USA
[6]Phycoil Biotechnology International, Inc., Fremont, CA 94538, USA
[7]Bioengineering Program, Biological and Environmental Sciences and Engineering Division, King Abdullah University of Science and Technology (KAUST), Thuwal 23955-6900, Kingdom of Saudi Arabia
*Correspondence: kyle.lauersen@kaust.edu.sa

## SUMMARY

Algae are diverse organisms with significant biotechnological potential for resource circularity. Taking inspiration from fermentative microbes, engineering algal genomes holds promise to broadly expand their application ranges. Advances in genome sequencing with improvements in DNA synthesis and delivery techniques are enabling customized molecular tool development to confer advanced traits to algae. Efforts to redesign and rebuild entire genomes to create fit-for-purpose organisms currently being explored in heterotrophic prokaryotes and eukaryotic microbes could also be applied to photosynthetic algae. Future algal genome engineering will enhance yields of native products and permit the expression of complex biochemical pathways to produce novel metabolites from sustainable inputs. We present a historical perspective on advances in engineering algae, discuss the requisite genetic traits to enable algal genome optimization, take inspiration from whole-genome engineering efforts in other microbes for algal systems, and present candidate algal species in the context of these engineering goals.

## INTRODUCTION

### Algae: A disparate grouping of unrelated protists

Algae represent a broadly distributed and diverse group of organisms, ranging from large multicellular species resembling plants, such as kelp, to simple unicellular protists, such as picoeukaryotes. Despite their shared ancestral origins, algae exhibit ecological, genetic, and physiological diversity that exceeds that of most other taxa.[1] In general, algae have plastids that contain remnant genomes originating from a singular primary endosymbiotic event involving uptake of a cyanobacterium by a eukaryotic protist. Most, but not all, algae retain the capacity for photosynthesis. After the initial endosymbiotic event, three main groups of algae arose and served as the foundation for what are now the dominant photosynthetic eukaryotes, including land plants.[2] Although the diversity of algae extends to secondary, tertiary, and even quaternary endosymbiotic groups of organisms,[3] we will discuss only primary endosymbiotic algae belonging to the red, green, and Glaucophyceae groupings that exist within the plant kingdom.

### Red, green, and Glaucophyta algae

Rhodophyta (red), Chlorophyta (green), and Glaucophyta are the three primary lineages of algae.[4] Rhodophyta and Glaucophyta retain phycobilisomes, which are ancestral cyanobacterial light-harvesting structures, with Rhodophyta using both red phy-

coerythrobilin and blue phycocyanobilin pigments for light capture, while Glaucophyta only employ blue phycobilins.[5,6] The Chlorophyta perform light-harvesting using chlorophyll *a* and *b* within plant-like chloroplasts. The blue-green-hued Glaucophyta plastids, called cyanelles, retain bacterial-like peptidoglycan walls,[7] which are found also in some chlorophyta, byrophytes, and tracheophytes.[8] Photosynthetic eukaryotes evolved a single-step biosynthesis pathway for the most prevalent plastid lipid, monogalactosyldiacylglycerol, compared to a two-step process in the cyanobacterial progenitors.[9] Starch storage is another shared trait that evolved in eukaryotic algae.[6]

Algae can be used in biotechnology for bioconversion of low-value inputs, such as carbon dioxide, nitrogen/phosphorus streams, and trace elements often found in wastewaters, into biomass and higher-value bioproducts (Figure 1). Photosynthetic algal cultures require only minimal chemical inputs compared to fermentative organisms that rely on reduced carbon sources for growth, and many algae also thrive under mixotrophic conditions, making them ideal candidates for upcycling complex wastewaters.[10] Examples of algae-derived products include nutraceuticals made from whole-cell biomass or carotenoid-enriched extracts of unmodified *Dunaliella salina* and *Haematococcus lacustris* (formerly *pluvialis*). The Trebouxiophyte alga *Chlorella vulgaris* has GRAS (generally recognized as safe) status and has been cultivated as an alternative food for decades.[11] Several companies are looking at classically mutated

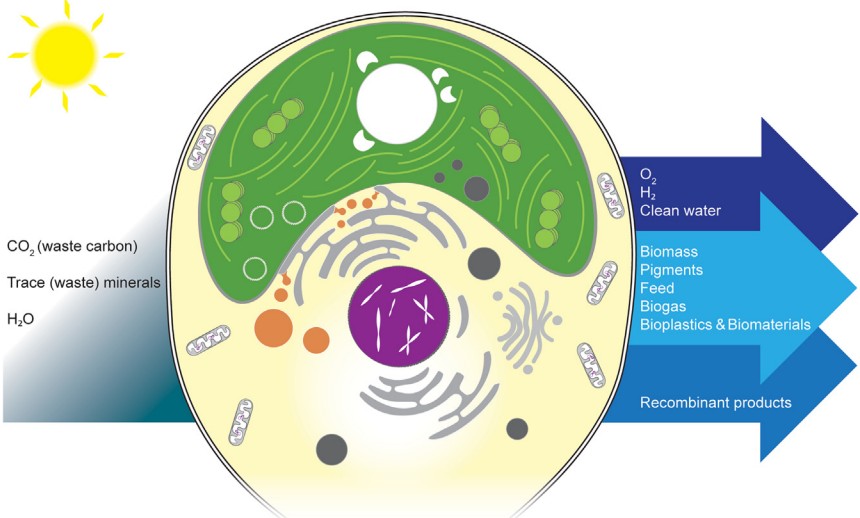

**Figure 1. The algal cell is a platform for harnessing energy from photosynthesis to convert basic inputs into more complex chemicals**

A simplified cellular architecture, illustrated here, shows a primary endosymbiotic alga containing a plastid that was derived from uptake of a cyanobacterium by an ancestral protist. Genomes are present in the nucleus (purple), mitochondria (light gray and white), and plastid (green). Various other subcellular compartments characteristic of eukaryotes are illustrated, including the ER (gray cisternae), Golgi apparatus (light-gray cisternae), lipid droplets (orange), and vacuoles or other microbodies (dark gray). Eukaryotic algae accumulate starch for carbon storage (white). In addition to photosynthetic production of oxygen ($O_2$), some species can generate molecular hydrogen ($H_2$) under specific conditions. Algae transform water, $CO_2$, and inorganic nutrients into biomass containing valuable molecules with diverse applications. Efficient uptake of dissolved nutrients can mitigate eutrophication and yield clean water. Genetic engineering of algal genomes opens possibilities beyond natural products to generate valuable biochemicals and recombinant proteins from minimal inorganic inputs.

variants of green algae with reduced pigments and fermentative growth to use their biomass as single-cell protein for human food (Triton Algae Innovations [https://tritonai.com] and Algenuity [https://www.algenuity.com]). Rhodophyta have long been used for food and agar, a natural hydrocolloid that is a fundamental product for microbiology.[12]

Algal biotechnology mainly focuses on unicellular or multicellular microbial species cultivated in photobioreactors of various dimensions or open pond systems.[11] Enclosed systems such as photobioreactors and fermentors offer superior control over culture conditions, invasive species, and pests but are more expensive than open (pond) systems.[13] Algae, in open or closed photobioreactors, can be cultivated on non-arable land using brackish or salt water and can have higher theoretical productivity in biomass per unit time than traditional land crops.[14–16]

### Opportunities for engineering algae

Beyond fundamental research, genetic engineering of algal genomes to yield alternative traits and novel products now provides opportunities to add additional layers of value to their cultivation. The promise of algal biotechnology theoretically lies in the use of photosynthetic microbes that can be grown in wastewater using solar energy to capture nutrients that would otherwise be discarded and convert them into specific chemicals of interest.

The organellar compartmentalization of algal cells provides a platform for the production of either non-glycosylated recombinant proteins in the chloroplast[17] or glycosylation of proteins in the endoplasmic reticulum (ER) and subsequent secretion.[18] Algal cells may also be favorable eukaryotic environments for metabolic engineering approaches to produce specialty metabolites by expressing vascular plant enzymes.[19,20] However, several barriers hinder the broad implementation of algal species as host cell systems for molecular engineering. Some species may be easy to cultivate but are not genetically tractable, thereby necessitating substantial effort to achieve genetic manipulation.

Photosynthetic growth, while advantageous from a sustainability perspective, also necessitates infrastructure considerations to ensure adequate light penetration and carbon dioxide supply into cultivation vessels, factors that generally yield lower volumetric cell densities than fermentation. Nevertheless, these challenges present opportunities, and several research groups have modified diverse species of algae to make various products, covered in multiple reviews: hydrogen and diesel biofuels,[21] fragrances, flavors, cosmetics,[19,22] and pharmaceuticals.[23,24] The key advantage algae bring to biotechnology is their capacity for photosynthesis and propensity for inorganic nutrient uptake, which enables them to use the energy from sunlight for the synthesis of complex chemicals from repurposed waste streams (illustrated in Figure 2). Native natural products can be sourced from algal biomass, and genetic engineering can be used to enhance yields of these products or generate non-native biochemicals in the algae.

Algal biomass can be separated into hydrophobic and hydrophilic fractions and may also include secreted products such as extracellular polysaccharides. Sourcing multiple products of value from a single cultivated organism is promoted as a "biorefinery" concept, akin to the fractionation of crude oil in a petroleum refinery. In this context, engineered algae could yield multiple native and engineered target products simultaneously. For example, our recent report showcased the potential of engineered algae grown in post-anaerobic-treatment wastewater to simultaneously generate biomass, purify the water of ammonium and phosphorus, remove $CO_2$, and produce a heterologous terpenoid.[10] Another report demonstrated genetic engineering for the concomitant production of ketocarotenoids in algal biomass and volatile isoprene hydrocarbons in the culture headspace.[32] These examples illustrate the potential of engineered algae as light-driven, multi-product-generating cellular platforms for waste-stream revalorization.

CellPress

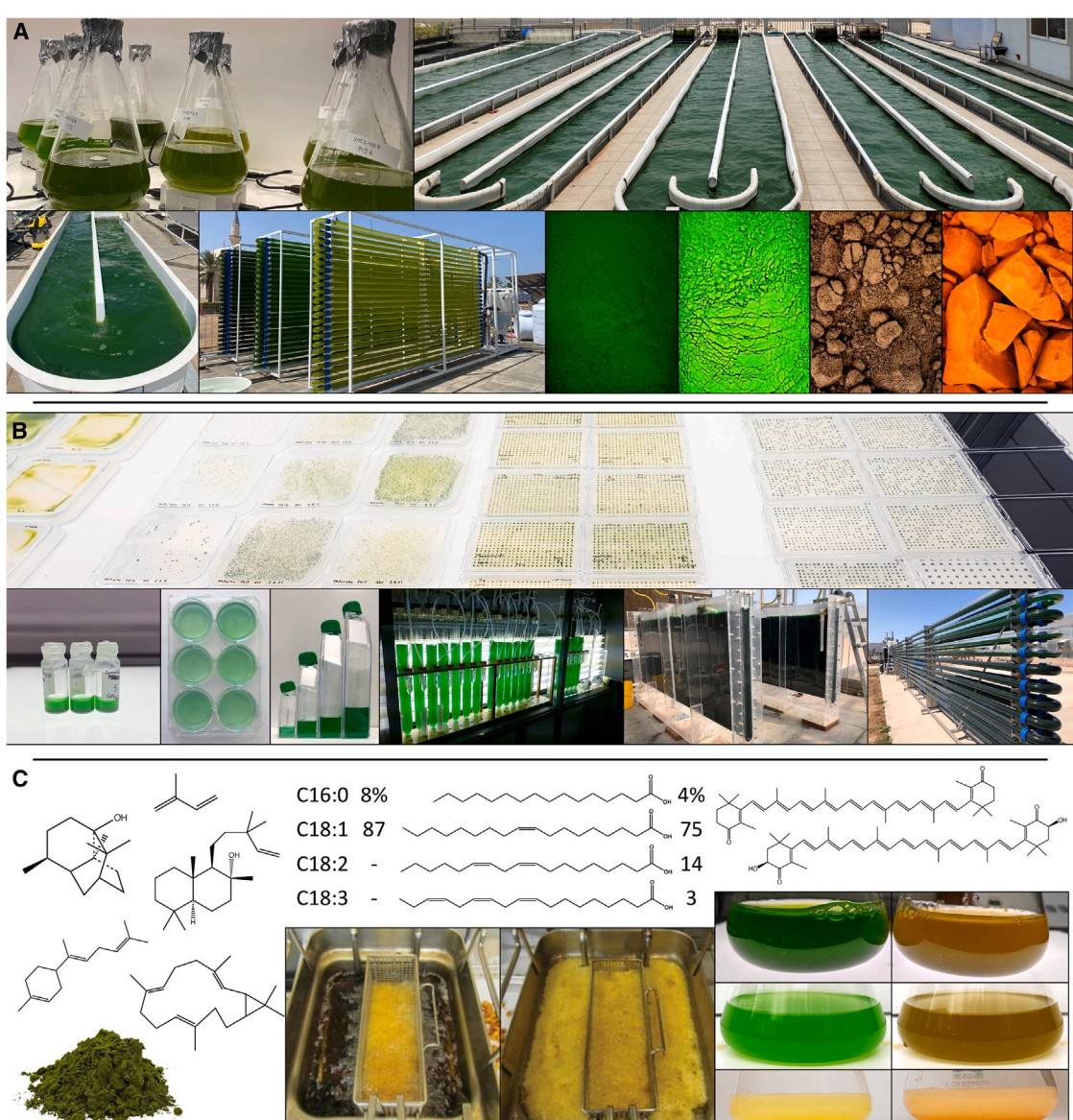

**Figure 2. Native and engineered algae for biotechnology**

(A) Algae cultures are scaled up from small volumes in flasks (upper left) to several thousand liters in raceway ponds (upper right) or tubular photobioreactors (lower middle). Photos of outdoor algal cultivation are of the Phase I facilities of Development of Algal Biotechnology in Kingdom of Saudi Arabia (DAB-KSA) – Beacon Development project funded by the Ministry of Environment Water and Agriculture and run by Dr. Claudio Fuentes Grünewald at King Abdullah University of Science & Technology (KAUST) Campus from 2023 (photo credit author K.J.L.). The product of algal cultivation is biomass with species-specific composition of pigments and other compounds; from left to right are dried green algae biomass from *Tetraselmis*, *Chlorella*, *Haematococcus*, and *Dunaliella* sourced from Algikey, Portugal, in 2022 (photo credit Sergio Gutiérrez).

(B) Engineering of algae involves the transformation of designer DNA, followed by the selection of transformant colonies and their subsequent isolation and characterization. The workflow for *C. reinhardtii* transformant recovery and high-throughput screening on agar plates is shown (top). Scale-up of engineered algae is the same as for wild-type strains, with increasing culture sizes providing the inoculum until production volumes are achieved (bottom) (pictures and images provided by Dr. Mark Seger of Arizona Center for Algae Technology & Innovation).

(C) Left: among other compounds, *C. reinhardtii* has been genetically modified to produce heterologous isoprenoids, including patchoulol,[25,26,27] sclareol,[28] bisabolene,[29] casbene,[30,31] and volatile isoprene.[32] Middle: *P. moriformis* was engineered to make tailored triacylglycerol oils. The images show a high-stability oil (left panel) that remains unfouled after 10 days of continuous deep frying, compared to high-oleic Canola oil (right panel). The relative percentages of major fatty acids in the oils are indicated above the pictures (reproduced from INFORM magazine,[33] with permission). Right: *C. reinhardtii* (top[34,35]), *C. merolae* (middle[36]), and *A. protothecoides* (bottom[37]) have all been engineered to produce orange-red ketocarotenoids such as canthaxanthin and astaxanthin (structures shown above) using the *C. reinhardtii* β-carotene ketolase 1 (*BKT1*). Wild-type strains are on the left, and modified strains are on the right. The *C. reinhardtii* pictures were provided by Dr. Thomas Baier and Jacop Kneip, Universität Bielefeld. The *C. merolae* pictures are from Seger et al.[36] *A. protothecoides* (photo by author J.L.M.) was grown heterotrophically on organic carbon and is non-photosynthetic in this state. The yellow pigmentation in wild-type *Auxenochlorella* is primarily from lutein.

Reliable and advanced genetic modification has been confined to a limited number of model algae (examples in Figure 2). However, improving knowledge of newly characterized species with favorable genetic and growth characteristics is expanding the repertoire of algae that are suitable for engineering approaches. This has been facilitated by the advent of cost-effective whole-genome sequencing and synthesis of custom DNA molecular tools for genetic modifications in emerging species. As markets and product demands constantly evolve, this review refrains from endorsing specific biochemical targets for algal engineering. Rather, we will emphasize features of importance and the potential value of using algae as platforms for synthetic biology and metabolic and genome engineering, as well as for basic discovery. Advances in genome manipulation, both for small-scale targeted engineering and more comprehensive whole-genome redesign, will play pivotal roles in defining future applications for these diverse organisms.

## KEY DEVELOPMENTS IN ENGINEERING ALGAL GENOMES

### Algae that are genetically tractable model organisms

Classical genetics in algae has been instrumental in generating auxotrophic strains and dissecting essential pathways since the 1960s.[38] These approaches were primarily focused on the unicellular Chlorophyta, *Chlamydomonas reinhardtii*, which continues to be the premier model organism for genetic, biochemical, and biophysical studies of photosynthesis and eukaryotic cilia/flagella.[39] The nuclear and organellar genomes have all been transformed,[40–42] and the development of improved markers and gene expression toolkits is ongoing.[43,44] However, the availability of genomic data for other algae is rapidly increasing, facilitating the design of gene expression cassettes for genetic interrogation of emerging species.[1,45]

These datasets have enabled the development of molecular tools for nuclear transformation of a number of red and green algae, although successful transformation of Glaucophyta has not yet been reported.[16] Examples of species in which new molecular tools have enabled transformation include *Ostreococcus tauri*,[46] *Auxenochlorella protothecoides*,[37] and *Galdieria partida*.[47] Transformation of *Volvox carteri* has also been demonstrated for fundamental analyses.[48,49] Molecular genetic tools have been developed for *Ulva mutabilis*, the first multicellular green alga besides *V. carteri* to be amenable to biotechnological exploitation.[50] However, transformation of some algae is inconsistent or difficult to reproduce or has not gained traction with a broad community of researchers, e.g., *Chromochloris zofingensis*[51] and numerous *Chlorella* spp.[52]

In addition to nuclear genome engineering, plastid genome engineering in algae is possible, and even genetic transplantation of whole plastid genomes between species has been reported.[53] In addition to *C. reinhardtii*, the chloroplast genome of the red alga *Cyanidioschyzon merolae* 10D has been transformed,[36,54] as has the fast-growing green *Picochlorum* spp.[55] (discussed later). Much remains to be understood regarding transgene expression in algae, and there are numerous species-specific features that require consideration for each organism in which engineering is a goal.

### *Chlamydomonas reinhardtii* as a long-standing model alga

*C. reinhardtii* has served as the prototypical green alga for laboratory investigation since its first isolation in Amherst, USA, in 1945 and remains the most extensively studied algal species to date. It has a division time of 6–8 h; can grow photoautotrophically, photoheterotrophically, or heterotrophically on media containing acetate as a fixed carbon source; and is easy to isolate into axenic cultures. With chloroplasts similar to those of plants, *C. reinhardtii* has been adopted as a model for chlorophyceae photosynthesis. Beyond that it has been used to study starch biosynthesis, circadian rhythms, microtubule formation and flagellar movement, and sexual reproduction, and the light-gated ion channels discovered in *Chlamydomonas* have opened up the field of optogenetics.[56]

Transformation of DNA into *C. reinhardtii* can be achieved by agitation in the presence of glass beads, biolistic bombardment with DNA-coated nanoparticles, or electroporation.[57] As is typical of primary endosymbiotic algae and plants, *C. reinhardtii* cells maintain separate nuclear and plastid genomes, each harboring unique features and potentials for genetic engineering.

### The non-nuclear genomes of *C. reinhardtii*

The plastid genome of *C. reinhardtii* is evolutionarily linked to those of modern-day plants.[2] Plastid compartments are a distinct microenvironment from the cytoplasm, and most of the proteins that function in the plastid are nuclear-encoded, translated on cytoplasmic ribosomes, and imported into the organelle.[58] The chloroplast is more suitable for biotechnological applications such as recombinant protein overexpression than the mitochondrial genome.[42,59] The chloroplast genome copy number ranges from 40 to 100,[60] requiring selection pressure to achieve homoplasmy, a state in which all genome copies are the same, following a transformation event. However, this copy number can facilitate high levels of recombinant protein expression. The chloroplast genome has a strong A/T codon bias compared to the nuclear genome, and transformed DNA is integrated by homologous recombination (HR), a feature that enables precise targeting of transgene insertion.[58] The *C. reinhardtii* plastid can function as a compartment for recombinant protein accumulation, and it has been shown that disulfide bond formation as well as proper folding of complex non-native proteins can occur within the organelle.[61,62]

The *C. reinhardtii* plastid genome has been proposed as a miniature test bed for synthetic genomics.[63,64] Codon reassignment was demonstrated here, taking advantage of the absence of the TGA stop codon in any coding sequence. A tryptophan tRNA was recoded to recognize UGA, enabling the plastid expression of recombinant proteins that cannot be translated in other hosts.[65] The use of antibiotic markers in plastid transformation can be avoided by using native genes to complement mutants with disrupted plastid-encoded photosynthetic genes.[66,67] Plastid expression of the *Pseudomonas stutzeri* WM88 phosphite oxidoreductase (*ptxd*) gene was shown to enable use of inorganic phosphite as a phosphorus source. This metabolic enhancement has dual functions as a transformation marker and in reducing contamination by other organisms that cannot metabolize phosphite.[68,69] Numerous recombinant proteins have been produced in this organelle, and iterative

transformations have been made possible by the development of chloroplast selection marker recycling methods.[70]

The plastid genome of *C. reinhardtii* will likely be the first to be synthetically redesigned, as efforts have commenced to completely recode this in several research groups. At the time of writing, *C. reinhardtii* plastid genome engineering projects are under way in the groups of Saul Purton (University College London), Alison Smith and Jason Chin (Cambridge),[71] Tobias Erb (Max-Planck-Institute for Terrestrial Microbiology, Marburg), Duanmu Deqiang (Huazhong Agricultural University, Wuhan), and Zhangli Hu, Chaogang Wang, and Bin Jia (Shenzhen University).[72]

### Major milestones in the development of C. reinhardtii nuclear genome transformation

*Chlamydomonas* was one of the first algae to have a high-quality genome sequence publicly available.[73] Initially, forward genetics screens were the focus of *C. reinhardtii* research. Early chemical and radiation mutagenesis studies generated cell-wall-deficient strains,[74] starchless mutants,[75] strains used in uncovering the xanthophyll cycle and its role in photoprotection,[76] and lipid biosynthesis mutants,[77,78] among many others. A key development in *C. reinhardtii* genetic engineering was the glass-bead-mediated transformation protocol, which enabled random integration of DNA into the nuclear genome of cell-wall-deficient or -removed strains,[40] thereby enabling some expression of transgenes. Transformation and insertional mutagenesis were initially achieved using endogenous genomic or cDNA sequences to complement auxotrophs, strains lacking a native metabolic capacity.[79] However, achieving reliable transgene expression was previously a challenge.

Some antibiotic resistance markers that could be employed in early *C. reinhardtii* transformation experiments included a *Streptoalloteichus hindustanus* (*Shble*)[80,81] gene for bleomycin family antibiotic resistance and the high-GC-content *Streptomyces hygroscopicus aphVII*[82] and *S. rimosus aphVIII*[83] genes for hygromycin and paromomycin resistance, respectively.[81,83] These studies revealed the importance of codon optimization and selecting appropriate endogenous promoters and terminators in *C. reinhardtii* gene expression cassette design.[84] It was demonstrated that inclusion of the first intron of the *Ribulose-1,5 Bisphosphate Carboxylase/Oxygenase Small Subunit 2* (*RBCS2*) gene improved the expression of antibiotic resistance markers,[80,85] as well as luciferase and fluorescent reporters,[85–87] and substantial improvements in nuclear transgene overexpression were achieved through the repetitive spreading of this intron throughout synthetically designed transgenes.[25,88] The strong tendency of *Chlamydomonas* to inactivate transgenic DNA sequences during integration into the nuclear genome has also been mitigated by the development of domesticated strains with mutations in an Sir2-type histone deacetylase involved in epigenetic gene silencing.[89,90]

The *Chlamydomonas* synthetic biology toolkit now includes an almost gapless 114-Mb nuclear genome assembly,[91] multiple transformation markers,[92,93] defined parameters for transgene expression,[94] localization signals that can be used to direct proteins to specific subcellular compartments and fluorescent proteins tags for visualization,[95,96] and ribonucleoprotein (RNP)-mediated gene editing.[97–99] Libraries of knockout mu-

tants have also been generated for use by the research community,[100] and these developments combine to now enable sophisticated metabolic and genome engineering in *Chlamydomonas*. The community-developed modular cloning "MoClo" kit for *C. reinhardtii* continues to be a major collaborative effort toward standardizing *Chlamydomonas* engineering.[44] Key biotechnology applications using this alga include demonstrating synthesis of valuable commodity biochemicals, including terpenoids such as isoprene[32] and polyamines such as cadaverine.[101] These efforts illustrate the utility of engineered algae for producing tailored chemicals, particularly in the context of wastewater valorization.[10] Biotherapeutic applications include engineered biocontrol of pathogens in aquaculture[102] and expression of anti-cancer immunotoxins.[62]

### Future crucial genome engineering developments to enable use of C. reinhardtii as a chassis organism

The extensive expertise that has been accumulated in the field of *Chlamydomonas* biology makes this alga an attractive system for engineering purposes. However, there are significant challenges that must be overcome if it is to reach its potential as a platform for genome-level engineering. An important consideration is its relatively large genome size (114 Mbp,[91] Figure 3), which is currently beyond the scope for complete synthetic redesign. Additionally, transgene integration into the nuclear genome occurs primarily by random non-homologous end joining (NHEJ) rather than HR,[103] so making large-scale targeted DNA replacements is challenging. Deletion or downregulation of components of the NHEJ pathway for DNA repair such as *Ku80* increased the frequency of HR-mediated integration by up to 70% in *Kluyveromcyes marxianus* and other yeasts.[104] RNP-mediated gene editing to make knockouts and targeted insertions of small transgene cassettes has been demonstrated in *Chlamydomonas*,[98,105,106] and the frequency of homology-directed repair (HDR) in gene editing was improved by mutagenesis of *Ku80*.[107] However, targeted integration of large, complex, multi-gene expression constructs for synthetic biology has not yet been demonstrated.

Improvements in gene-editing protocols have shown promise for targeting integration of gene expression cassettes through HDR.[99] However, recycling of marker genes in the nuclear genome has not yet been reported, so the number of successive engineering events that can be performed is limited by the availability of selection markers. Automation of genetic engineering, facilitated by robotic handling[114] in conjunction with high-throughput analytics and phenomic screens,[115] will play a role in increasing the speed of algal strain development.[116] Older techniques, such as protoplast fusion, whereby two cells can be physically merged and aspects of their genomes combined, may be useful in engineered strains when mating or marker recycling (discussed in the next section) is not feasible.[117,118]

While *C. reinhardtii* is likely to maintain its position as the premier model green alga, its complex nuclear genome may be too large for total redesign engineering to generate a completely controllable cell chassis. Other algae, with simpler genome architectures and the ability to perform HR, may be better suited for such investigations and are discussed in the next section (Figure 3).

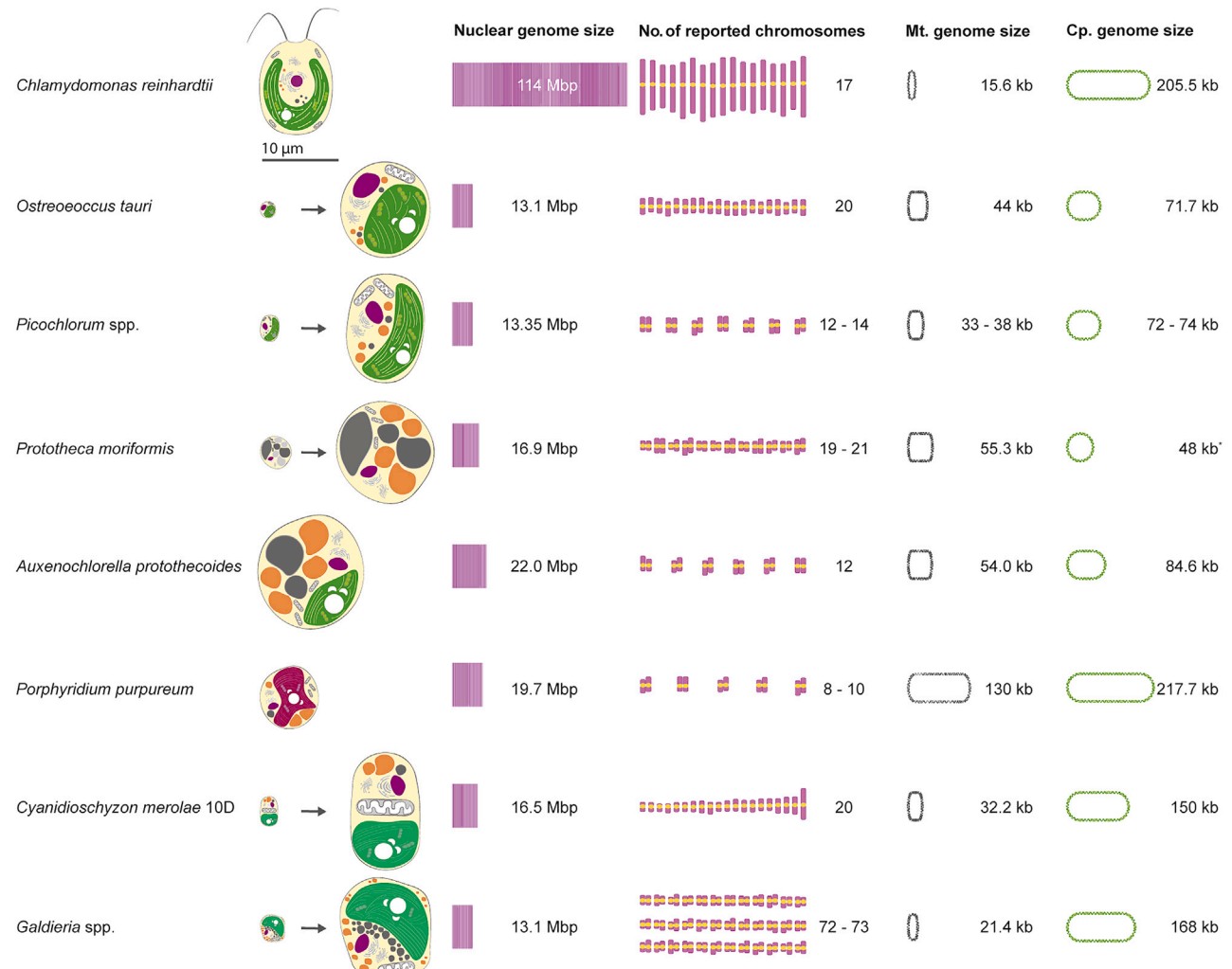

**Figure 3. Algal genomes that are candidates for engineering and resynthesis**
The nuclear genomes of the algae in this list are substantially smaller than the model Chlorophyta *C. reinhardtii*, making them more suitable for complete genome redesign. Cartoons representing each genus or species illustrate the relative cell sizes and pigmentation differences. The ploidy of the vegetative state is indicated by paired chromosomes for diploid and single chromosomes for haploid. The total haploid numbers of chromosomes in each nuclear genome, and the sizes of the mitochondrial and plastid genomes, are indicated. Genome data are from the following references: *C. reinhardtii*,[91] *O. tauri*,[108] *Picochlorum* spp.,[109] *Prototheca* spp.,[110] *A. protothecoides* (unpublished data), *P. purpureum*,[111] *C. merolae* 10D,[112] and *Galdieria* spp.[113] Plastids (green, except phycoerythrin-containing *P. purpureum* and heterotrophic *Prototheca* spp.), nuclei (purple), starch deposits (white), vacuoles and microbodies (gray), lipid bodies (orange), and mitochondria, ER, and Golgi (illustrated cartoons).

## THE POSSIBILITIES OF SYNTHETIC GENOMICS IN ALGAE

### Desirable traits to enable synthetic genome design in algae

Beyond the targeted disruption of genes and the insertion of cassettes for gene expression is the concept of whole-genome redesign; this strategy involves expanding functionality by systematically re-engineering and replacing entire chromosomes to eventually build a synthetic tailored genome. Essential aspects of redesign include removing redundancies in codon usage and thereby liberating tRNAs for recoding alternative amino acids, removing repetitive genome se-

quences, disentangling overlapping coding sequences, adding PCR tags to genes, incorporating specific recombination sites to enable whole-genome plasticity, or adding synthetic supernumerary neochromosomes—*de novo* chromosomes that are not found in nature—for specific purposes.[119,120] The Synthetic Yeast Genome Project (Sc2.0) (collected articles of which can be found at https://www.cell.com/consortium/synthetic-yeast-genome) is a global consortium effort to redesign the genome of this model organism, aiming to rebuild the 11.3-Mb *Saccharomyces cerevisiae* genome with many of these features. This project is replacing wild-type chromosomes with synthetic sequences to remove non-essential introns and tandem repeats in coding sequences, along with

long terminal repeats from viral DNA. Consolidated tRNAs are also being relocated to a neochromosome.

To leverage these technologies in algae and truly customize algal genomes, it will likely be only practical to focus on those with naturally small, non-redundant genomes that contain close-to-minimal gene sets. For organisms with larger genomes, such as *C. reinhardtii*, engineered rearrangements in native chromosomes may be facilitated if improvements in HDR can be realized.[121] However, genomes with lower total nucleotide footprints will be simpler to build *de novo*. Smaller genomes may have fewer hidden genetic elements, such as non-coding RNA within introns, or other uncharacterized genome structure features that may hinder engineering efforts.

The generation of redesigned genomes is currently realized in yeast by stepwise replacement of sections of native chromosomes with redesigned variants. For this purpose, HR of transformed DNA into the genome is essential, along with efficient counter-selectable transformation markers. Chromosome section replacement can be enacted by alternating between two selectable markers and stepwise walking across chromosome sections. A technique for this is called switching auxotrophies progressively for integration, as used in *S. cerevisiae*.[122] Marker genes such as *URA3*, encoding orotidine 5-phosphate decarboxylase, or *MAA7*, encoding tryptophan synthase β subunit, can be positively selected for restoration of uracil or tryptophan prototrophy, respectively. Excision of the markers (negative selection) restores auxotrophy and enables survival on 5-fluororotic acid or 5-fluoroanthranilate, which are otherwise converted into toxic products, leading to cell death.[123,124] Targeted modifications from CRISPR-Cas techniques can also assist in fine-tuning sequences within newly designed chromosome sections.[120,122,125]

Rapid growth is another key trait required to enable future genome engineering in algal platform strains. All genetic modification goals necessitate multiple rounds of transformation; therefore, a rapid doubling time is crucial to achieve genome-scale alterations within a reasonable time frame. The ability to grow heterotrophically can also be useful for the host cell, improving the likelihood of recovery after transformation events replacing large chromosomal sections, which can interrupt core cellular functions such as photosynthesis.[126] *C. reinhardtii* has been an advantageous model for teasing apart photosynthetic mechanisms, since non-photosynthetic mutants can be recovered on acetic acid.[57]

Haploidy can be another advantage, since genomic modifications can be more complex when there are two potential targets, but diploidy can also offer robustness and the ability to remove one copy of essential genes. Sexual reproduction can expedite the construction of partially synthetic chromosomes in tandem, and meiotic crossover can be exploited to assemble multiple parts into a full-length functional synthetic chromosome in a daughter cell.[127] Small genomes with high numbers of chromosomes can be advantageous for minimizing the sizes of synthetic chromosomes that have to be replaced and for consolidating genes for particular biochemical pathways onto specific chromosomes. Lower inherent intron density in the starting strain reduces the effort required to identify hidden regulatory elements that are contained in those features.

## Algae with natural features that hold potential for whole-genome engineering

### Ostreococcus spp.

*Ostreococcus* has emerged as a model picoeukaryote, particularly for investigations into the ecological niche-specific adaptations exhibited by four ecotypes or species—*O. tauri*, *O. lucimarinus*, *O. mediterraneus*, and *O.* sp RCC809, which were isolated from different marine environments[128]—and in studies of circadian rhythms.[129] *O. tauri* possesses a compact haploid genome and one of the smallest known chloroplast genomes (Figure 3), making it a good candidate for exploring minimal gene sets. Its nuclear genome is 12.56 Mb spanning 20 chromosomes, its mitochondrial genome is 44.3 kb, and its chloroplast genome is 71.7 kb.[108] Doubling times for *O. tauri* have been reported between 13.3 and 6 h.[130,131] Approximately 39% of genes are predicted to contain introns, which is considerably lower than the 92% in *C. reinhardtii* but significantly higher than the 5% of *S. cerevisiae*. *O. tauri* has been transformed by electroporation[132] and polyethylene glycol (PEG) treatment.[46] Efficient HR, requiring short homology arms for gene targeting, has been reported only once[133] and may require further characterization. Chromosomes 2 and 19 are particularly noteworthy for synthetic genomics applications, as they harbor the majority of its transposable elements and many genes encoding transporters, resembling prokaryotic genome islands that are enriched in genes acquired through horizontal gene transfer (HGT).[134]

### Picochlorum spp.

*Picochlorum* spp. represent a genus of high-light and thermotolerant, euryhaline picoeukaryotic Trebouxiophytes that have been isolated from a range of aquatic environments.[135] *Picochlorum* spp. have high photoautotrophic biomass productivity, with doubling times as short as 2 h reported for *P. celeri*.[136] Genome sequences are available for several *Picochlorum* spp.,[109] at least some of which are diploid, and the haploid sizes of their nuclear genomes range from 13 to 14 Mb (Figure 1).[137] *P. celeri* has a diploid genome of 27.43 Mb across 15 chromosomal contigs. Other species such as *P. soloecismus* and *P.* sp. SENEW3 have genomes as small as 13.5 Mb, with chloroplast and mitochondrial genomes of 72.7 kb and 38.8 kb, respectively. Improved thermotolerance in *P.* sp. *BPE23* was achieved through adaptive laboratory evolution and was associated with chromosome duplications.[138]

Electroporation and biolistics were used to transform the *P. renovo* nucleus and chloroplast, respectively,[55,139] and RNP-mediated gene editing enabled the targeted disruption of nitrate reductase and carotenoid isomerase genes in *P. celeri*.[140] Efficient secretion was driven by N-terminal fusions of native secretory peptides to fluorescent proteins in *P. renovo*, and this strain has been suggested as a platform for photoproduction of industrially relevant enzymes.[139] Development of molecular genetic tools is in an early stage for *Picochlorum*, but substantial progress has been made despite the relatively short duration of research.[137]

### Prototheca spp.

Solazyme/TerraVia Holdings produced "tailored oils"' for fuels, nutrition, and industrial applications by modifying fatty acid and lipid biosynthesis in the non-photosynthetic Trebouxiophyte, *Prototheca moriformis* UTEX 1435. These activities

represent probably the most advanced industrial use and genetic engineering of algae to date, yet key advances in manipulating *Prototheca* fatty acid/lipid composition are only reported in patent filings. *Prototheca* was selected as the industrial engineering platform on the basis of its robust fermentation performance, high lipid yield on glucose, and genetically tractable nature.[141] As is common for many oleaginous organisms, triacylglyceride (TAG) oil production in *Prototheca* was induced by nitrogen depletion in the presence of excess sugar.[141] Heterotrophic doubling time on glucose was reported to be 4 h.[142,143] A public genome sequence for *P. moriformis*, which is diploid,[141] is not yet available, but the haploid sizes of the nuclear genomes of related *Prototheca* spp. range between 16.7 and 20 Mb.[110,144] Although Prototheca spp. are obligate heterotrophs, they retain residual plastids, which are the site of bulk fatty acid biosynthesis.[110]

Genetic engineering in *Prototheca* was achieved by biolistic bombardment, and efficient nuclear gene targeting was found to occur by HR.[141] Selections for transformation included neomycin antibiotic resistance, rescue of thiamine auxotrophy, and glycosyl hydrolases that expanded the range of disaccharide sugar feedstocks suitable for fermentation.[141] *Prototheca* was also modified to express fungal pathways facilitating xylose uptake and conversion to xylulose, along with an *Arabidopsis* plastid transporter to enable incorporation of xylulose-5-phosphate into the pentose phosphate pathway. These modifications enabled xylose assimilation and enhanced utilization of cellulosic sugars in industrial fermentation.[145] Several additional metabolic engineering examples are highlighted in detail in the next paragraph to provide a cross-section of the approaches made possible by the genetic tractability of this alga. Currently, *Prototheca* stands as a prime example of the potential applications for genome engineering in algae; however, these species cannot be used in light-driven production systems.

*Prototheca* lipid is predominantly TAG, resembling olive oil, with palmitate (C16:0), oleate (C18:1n-9), and linoleate (C18:2n-6) as the predominant fatty acids. Strains accumulating high levels of C8:0, C10:0, C12:0, and C14:0 were engineered by overexpression of acyl-acyl carrier protein (ACP) thioesterases, β-ketoacyl-ACP synthases, and acyltransferases with substrate specificities toward medium-chain fatty acids (MCFAs).[146–148] High-oleic oils (>90%), with very low levels of saturated and polyunsaturated fatty acids, were generated by increasing elongation and stearoyl-ACP desaturase (SAD) activities in the chloroplast to enhance production of monounsaturated fatty acids, while downregulating the microsomal fatty acid desaturase FAD2, which converts oleate to linoleate.[149–151] Strains enriched with up to 70% stearate-oleate-stearate "structured fat" were created by expressing a C18:0-specific *Garcinia mangostana* FATA1 thioesterase, downregulating endogenous *SAD2* and *FAD2*, and co-expressing cocoa acyltransferases exhibiting a high degree of discrimination against esterification of saturated fatty acids at the *sn*-2 position.[152] *Prototheca* "algae butter" was compatible with cocoa butter and comparable in performance to shea stearin in confectionary applications.[153]

Fatty acid elongases (FAEs) from *Crambe abyssinica* and other species were used to modify the cytoplasmic elongation pathway, converting C18:1n-9 to C22:1n-9 (erucic acid), a fatty acid used in the manufacture of film plastics.[150] The efficiency of elongation was enhanced by Lands cycle acyltransferases, which increased the exchange of acyl groups between diacylglycerol, membrane phospholipids, and the acyl-coenzyme A (CoA) pool.[150] Up to 20% erucic acid content was achieved by expressing FAEs, upregulating homomeric acetyl-CoA carboxylase, and co-expressing β-ketoacyl-CoA reductase, 3-hydroxyacyl-CoA dehydratase, and enoyl-CoA reductase transgenes.[150]

*Prototheca* was also employed as a platform for cannabinoid biosynthesis by Purissima.[154] Enzymes from *Cannabis sativa* to make olivetolic acid were co-expressed with prenyl transferases and cannabinoid synthase genes, and accumulation of olivetolic acid, cannibigerolic acid, cannabidiolic acid, and tetrahydrocannabinolic acid in *Prototheca* oil was confirmed by chromatography and mass spectrometry.[154] These examples of *Prototheca* metabolic engineering for commercial applications illustrate the possibilities of novel products achievable when an alga exhibits transformability, gene targeting by HR in the nuclear genome, and stable transgene expression. These features are requisite for future genome-scale manipulations.

### Auxenochlorella prototothecoides

*Auxenochlorella prototothecoides* is the closest photosynthetic relative to *Prototheca* spp.,[110] and thus far all of the engineering capabilities developed for *Prototheca* can be applied to *Auxenochlorella*.[37] Doubling times from 6 h heterotrophically on glucose to 16 h photoautotrophically have been reported.[155,156] Photoautotrophic lipid accumulation of up to 40% biomass has been reported,[157] and mixotrophic degradation and use of cellulosic material as a substrate for growth has been observed.[158] In contrast to *Prototheca*, which are obligate heterotrophs, *Auxenochlorella* retains the capacity for photosynthesis but suppresses photosynthetic complex accumulation and switches to heterotrophic growth in the presence of organic carbon sources.[159]

Transformation of *A. prototothecoides* UTEX 250 using a lithium acetate/PEG procedure and efficient gene targeting by HR were reported by Phycoil Biotechnology International.[37] Successful transformation and nuclear gene targeting are possible in two additional genetically distinct *A. prototothecoides* strains, UTEX 25 and 2341, and high-quality, gapless nuclear, and organellar genome assemblies of *A. prototothecoides* UTEX 250 have been generated (Figure 3) (unpublished data). The transformation markers developed for use in *Prototheca* were found to also function in *A. prototothecoides*, and both heterologous and endogenous promoters have proven effective in driving transgene expression.[37]

Engineering approaches to improve the value of *Auxenochlorella* biomass include disrupting one or both alleles of *lycopene cyclase epsilon* (*LCYE*) to change lutein/zeaxanthin ratios with concomitant expression of the *C. reinhardtii* β-carotene ketolase (*BKT1*), leading to the accumulation of oils containing varying mixtures of orange-red ketocarotenoids (Figure 2).[37] These engineering efforts were inspired by the demonstration of modified ketocarotenoid profiles in *C. reinhardtii* with *BKT1* recoding and overexpression.[34,35,160,161] Strains with increased polyunsaturated fatty acids and reduced ω-6/ω-3 ratios were

generated by knockin of strong promoters at the *FAD3* locus, accompanied by the expression of *Arabidopsis PDCT* and *Linum usitatissimum* (flax) *FAD3* transgenes.[37] Disruption of both alleles of squalene epoxidase (*SQE*) yielded strains accumulating over 1,000 ppm of squalene, comparable to the content of olive oil.[37] Co-expression of the FATB2 thioesterase and KASA1 β-ketoacyl-ACP synthase from *Cuphea wrightii* in *Auxenochlorella* led to accumulation of up to 35% MCFA (unpublished data). These results suggest that *A. protothecoides* is equivalent to *P. moriformis* as a platform for lipid and isoprenoid pathway engineering, with the added advantage of phototrophic or mixotrophic growth.[162]

*Prototheca* and *Auxenochlorella* have significant potential as platforms for applied genetic and metabolic engineering, but their suitability for complete genome redesign remains to be demonstrated. Diploidy may facilitate complex chromosome rearrangements by providing redundancy, but it may also pose challenges in achieving complete synthetic replacements. These species originally found traction in industrial sectors with limited knowledge sharing and are only now emerging as model organisms for academic research with publicly available genome and systems data.[158,163,164] Facile HR, robust heterotrophic growth, and their oleaginous natures make them attractive hosts for future genome engineering approaches.

### Porphyridium purpureum

The red alga *Porphyridium purpureum* is noteworthy for its genetic flexibility, having acquired up to 9% of its genome by HGT[111]; an implication of this could be that *P. purpureum* may have a natural propensity for production of foreign proteins. Indeed, bacterial origins of replication enabled plasmid vectors to be maintained in the *P. purpureum* nucleus as stable episomes, and expressed transgenes can accumulate up to 5% of total soluble protein.[165] Circular plasmids were transformed by biolistic bombardment, and the efficiency of co-transformation with 1:1 mixtures of two plasmids was 100%.[165]

HR in the *Porphyridium* nuclear genome has not been observed, but gene editing with CRISPR-Cas9 RNP was used to knock out the chlorophyll *a* synthase gene, resulting in strains with enhanced phycoerythrin production.[166] Transformation and RNP-mediated HDR are fundamental tools that may enable genome engineering approaches in *P. purpureum*. As illustrated in Figure 3, the *P. purpureum* nuclear genome is relatively compact at 19.7 Mb, with 8,355 predicted genes.[111] There are no reports describing a *Porphyridium* sexual cycle, but genes that are essential for meiosis are present in the genome.[111] *Porphyridium* growth can be enhanced by supplementation with some organic carbon sources, but even under mixotrophy, doubling times are still over 2 days long compared to rapid growers such as *Picochlorum*.[167,168]

*Porphyridium* is an example of an organism that could be cultivated for biorefinery processing, since these species can produce large amounts of sulfated extracellular polysaccharides for cosmetic and pharmaceutical applications, phycobilins, which are of interest as biodegradable dyes, and the polyunsaturated fatty acids arachidonic acid and eicosapentaenoic acid.[169] Genome engineering could be employed to generate strains with improved capacities to synthesize these, and other, value-added natural products.

### Cyanidioschyzon merolae 10D

*C. merolae* 10D is a polyextremophilic red alga that is a model for investigating the unique mechanisms of photosynthesis at low pH and high temperatures. It is an obligate phototroph, originating from acidic hot springs and lacking a cell wall.[170] In contrast with other red algae, the members of the Cyanidiophyceae have lost phycoerythrin biosynthesis, resulting in a cyan-green coloration.[170,171] *C. merolae* exhibits a 9-h doubling time and can be cultivated at 42°C and pH 2 or lower.[172] These conditions can confer significant advantages for scale-up, as the energy requirements are lower for heating than cooling, and low pH is useful for mitigating contamination.[172]

Its 16.5-Mb haploid nuclear genome spans 20 chromosomes, with approximately 55% GC content, and it possesses 32-kb mitochondrial and 150-kb plastid genomes.[112] Only 0.5% of *C. merolae* nuclear genes contain introns.[170] Transformation into the nuclear genome is achieved using a PEG-mediated protocol, and 500-bp homology arms are sufficient for gene targeting by HR.[173] Transformation and maintenance of circular episomal plasmids has been observed by our group.

*C. merolae* strains have been engineered to enable heterotrophic growth by expressing sugar transporters,[174] for increased lipid yields with a cyanobacterial acyl-ACP reductase,[175] and for production of valuable ketocarotenoids.[36] The compact and intron-poor nuclear genome, along with its genetic tractability, metabolic flexibility, and adaptations to extreme growth conditions, positions *C. merolae* as a promising candidate for transgene overexpression and genome redesign.

### Galdieria spp.

*Galdieria* spp. are red algae that have been isolated from similarly extreme environments as *C. merolae*. Unique among the Cyanidiophyceae, they are facultative heterotrophs capable of using as many as 30 carbon sources while retaining the ability to grow photoautotrophically or mixotrophically. *Galdieria* spp. possess compact nuclear genomes ranging in size from 11 to 16 Mb, with 72–73 chromosomes depending on the species, and about 45% GC content, despite occupying an environmental niche similar to that of *C. merolae*.[113,176] Approximately 1% of the genome was acquired by HGT.[176]

*Galdieria* display many features that would be beneficial for genome engineering; *G. partita* was discovered to have a sexual cycle, with meiotic conversion of cell-walled, vegetative diploid cells into motile cell-wall-less haploid gametes under pH 1, which can mate to produce diploid progeny.[47] Meiotic crossovers can facilitate the stepwise assembly of synthetic chromosomes or transfer of transgenes.

Haploid cells of *G. partita* have been transformed successfully by adapting the *C. merolae* PEG-mediated protocol; gene targeting by HR was reported to occur efficiently, as was marker recycling.[47] The only drawback to engineering *Galdieria* is its relatively slow 16-h doubling time,[177,178] but this may be mitigated by faster growth under mixotrophic conditions.[179] To date, only one study has demonstrated genetic manipulation in this genus, so continuing efforts are warranted to explore its potential as a synthetic biology platform.

### Novel genera

Bioprospecting for novel species is another avenue for discovering algae with the appropriate traits for genome

engineering. New algae species worthy of further investigation for biotechnological applications should grow rapidly, have relatively small, genetically tractable genomes, preferably be capable of efficient HR, and have high flux to valuable natural products.[180]

*Medakamo hakoo*, an ultrasmall (1 μm diameter) Trebouxiophyte discovered in an aquarium in 2015, is an example of a promising new algae species, possessing a compact 15-Mb nuclear genome and unusual cell division involving generation of four daughter cells per cell cycle. *M. hakoo* is in a sister clade to *Botryococcus* and despite its small size may have the potential to produce starch and high titers of TAGs. These natural features warrant further investigation, and successful transformation could open avenues for the use of this species for synthetic biology and biotechnology.[181]

## BROAD PERSPECTIVES OF SYNTHETIC GENOMICS AND THE PARADIGM OF WHOLE-GENOME APPROACHES TO ENGINEERING

### From pathways to genomes

Chromosome- and genome-scale engineering approaches in other organisms have been made possible by advances in DNA sequencing, DNA synthesis, and transformation of large customized DNA fragments.[182] Algal genome engineering will for some time likely remain limited to the introduction of short biochemical pathways or to protein overexpression. The plastid presents a logical starting point for synthetic genome redesign, and partial replacement of nuclear chromosome sections may be possible before whole synthetic chromosome replacements become a reality.[121] Neochromosomes containing multi-gene pathways in addition to RNP-mediated editing of the genome could enhance algae with valuable new traits or products before complete genomic redesigns are realized. In yeast, 56 genetic changes in total were employed to yield strains producing monoterpene indole alkaloids, combining 30 heterologous genes and multiple genome edits but without a synthetic genomics approach incorporating partial or complete chromosome redesign.[183]

To facilitate neochromosome construction for algal applications, techniques such as the "telomerator" could be employed. This is a seed sequence used to add telomeric caps to small DNA fragments to enable the creation of synthetic small chromosomes.[184] Neochromosomes were employed in yeast to encode a humanized version of the adenine biosynthesis pathway[185] and express an array of pan-genomic elements from industrial or environmental strains for evaluation of phenotypic diversity.[186] A 190-kb neochromosome relocated all 275 yeast tRNA genes to reduce the genomic instability associated with tRNA elements.[119]

The potential for completely re-engineering algal genomes is of interest to develop future cell lines that function more like programmable biological machines. Engineered algae of the future with entirely recoded genomes will be a chassis for light-driven bioproduction and waste circularization concepts, with a range of features that make them controllable and elegant to modify. Some of these potential features are illustrated in Figure 4.

### Lessons from synthetic genomics in relation to algae
*Biocontainment, genome architecture engineering, and synthetic speciation*

JCVI-Syn1.0 was the first example of an entirely synthetic genome and its successful transplantation into a restriction-deficient strain of *Mycoplasma mycoides*.[187] This demonstrated that a chemically synthesized copy of a genome, with significant modifications, could be used to replace a native genome. Among other features, watermarks and defined sequence tags were included, features that influenced latter genetic designs in more complex organisms. The current stage of the synthetic mycoplasma genome, JCVI-Syn3a, has shown that minimization of genomes can reduce redundancy and complexity and increase the predictability of a biological system. The genetic dissection of highly redundant gene families can unveil otherwise cryptic functions in genome stability that might be otherwise overlooked in the native genome context; analysis in a minimal genome is a useful way to fully characterize individual gene function of highly redundant genes.[188] This minimization approach, applied to algal genomes, could elucidate the roles of many genes that have only been assigned functions based on predictive annotation or models.

Within the Sc2.0 project, one included redesign feature is the placement of numerous *loxP* sites in terminator regions of coding sequences throughout the engineered genome.[189] Through inducible expression of the Cre recombinase, recombination between intergenic symmetrical lox sites (lox-*p*-sym) stimulates genome-wide terminator swapping, deletions, inversions, and duplications as well as translocations. However, unlike mutagenesis methods, this inducible randomization generally does not affect gene-coding sequences.[189] This technique is called SCRaMbLE (synthetic chromosome rearrangement and modification by lox-p-mediated evolution) and can be used to investigate rapid evolution of yeast strains with improved features such as tolerance to high pH[190] and heterologous metabolite production.[191] Similar features could be incorporated into future synthetic algal genomes to facilitate directed evolution, metabolic engineering, and reverse engineering or to improve strain growth performance.

Biocontainment becomes increasingly important as engineered organisms are deployed. Synthetic transgenes could be unwittingly introduced into the environment through outbreeding and HGT, risking adverse ecological outcomes. A number of mechanisms have been proposed to mitigate these risks, including codon reassignment and synthetic auxotrophy through dependence on non-natural amino acids.[192] Conditional suicide systems and tox/antitox gene pairs can also create boundaries for genetic escape.[193,194] *Salmonella typhimurium* and *Escherichia coli* strains expressing modified tRNAs to recode codon usage were resistant to viral infection and incompatible with bacterial HGT.[192] Similar approaches could be applied for crop protection and biocontainment in algae with engineered genomes.

Another approach to minimize genetic escape involves karyotype engineering to limit mating compatibility with wild-type strains. Circularization of a synthetic *S. cerevisiae* chromosome V demonstrated the flexibility of eukaryotic cells to function normally with uncommon chromosomal arrangements and the feasibility of more complex genome architectural design approaches.[195] Reproductively isolated *S. cerevisiae* strains where

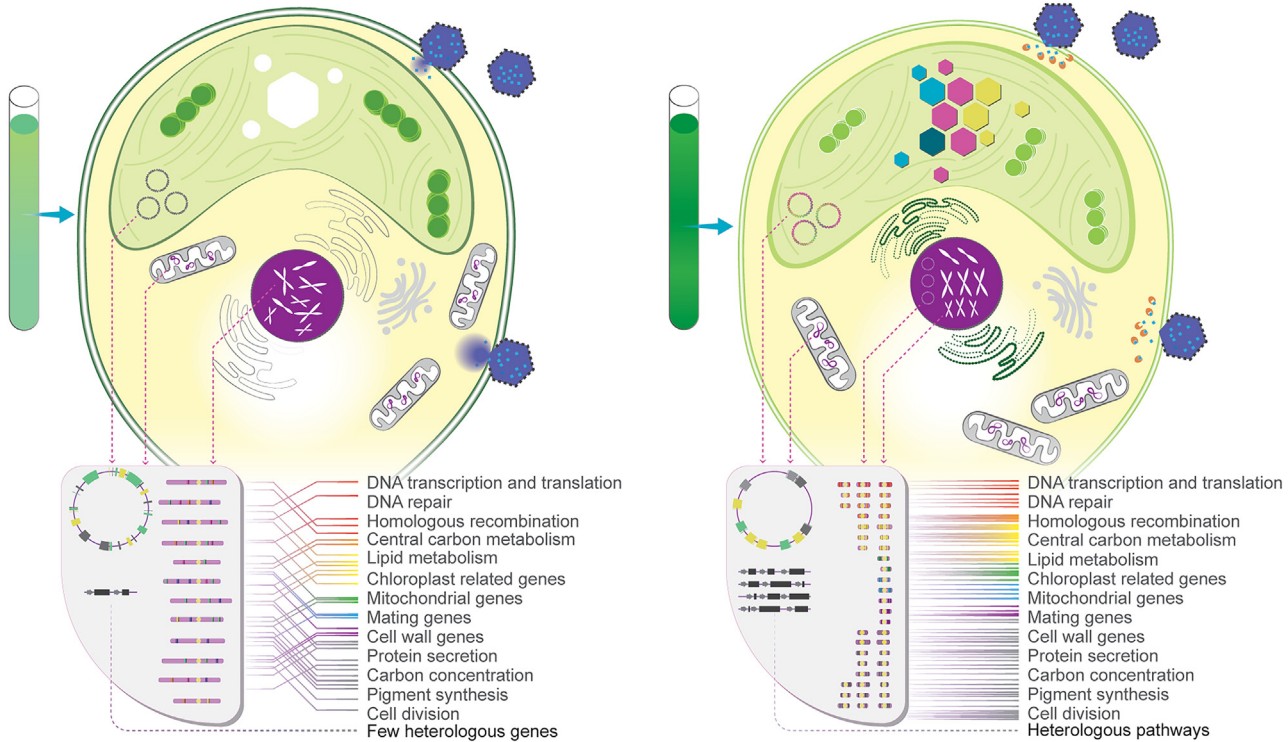

**Figure 4. The wild-type alga compared to the genome-engineered alga of the future**

Left: the wild-type alga maintains three separate genomes in the plastid, mitochondria, and nucleus. The nuclear genome may be haploid or diploid in vegetative cells, and genes for all cellular processes are distributed throughout the chromosomes. Genetic loci on any of the genomes may be modified by gene targeting and editing, and transgenes can be integrated and expressed. Simple heterologous pathways can be introduced, but broad genomic alterations, multi-locus targeting, and reliable expression of multiple transgenes of complex biochemical pathways is still challenging. The wild-type or domesticated alga is susceptible to pathogens, represented by blue hexagons, and accumulates both native and engineered products through synthetic biology. Right: the engineered alga of the future will feature completely recoded and redesigned genomes. The nuclear genome, either haploid or diploid, will be dispersed across numerous small autonomously replicating chromosomes, neochromosomes, or episomes. Plastid genomes may be rearranged, with additional gene transfer to the nucleus when advantageous. Redundancies and repetitions in the genome will be eliminated except where necessary to increase gene copy numbers for improved protein expression. Genes will be grouped into neighborhoods according to function to facilitate modular engineering, and recombination elements will be inserted to enable broad-scale genome shuffling on demand. The recoded algal genomes will be intrinsically resistant to pathogen attack and will encode defenses against competitors and grazers, along with biocontainment strategies such as engineered auxotrophy or mating incompatibility. Genome landing pads will streamline the integration of multi-transgene pathways and accelerate the biosynthesis of novel products, symbolized by colored hexagons in the plastid.

the whole genome is merged into one chromosome[196] were also created by RNP-mediated gene editing, but it should be noted that the inability to mate with wild type does not necessarily prevent genetic transfer into the environment. Algal species, such as those featured in Figure 3, may be interesting for such approaches, with those containing small genomes and performing HR being specifically promising candidates.

## FINAL NOTES

The future of algal synthetic biology offers hope for the efficient photosynthetic conversion of nutrients from waste streams into value-added biochemicals. Drawing inspiration from engineered genomes of bacteria and yeasts, we now have the foundational tools of high-quality algal genome assemblies, efficient transformation, some species with efficient HR, and maturing RNP-mediated gene editing. These tools, applied in hosts with favorable molecular genetic features, will enable genome-scale modification and redesign in several algae species. The seven algal genera detailed in "algae with natural features that hold potential for whole-genome engineering" are candidates for additional development, but bioprospecting efforts should continue to seek fast-growing, genetically tractable species with superior traits for genome engineering. Genome minimization, chromosome architecture restructuring and refactoring, and the transplantation of metabolic pathways onto neochromosomes are all becoming feasible approaches in algae systems and can be applied to genome-scale engineering projects that were once purely hypothetical.

## SUPPLEMENTAL INFORMATION

## ACKNOWLEDGMENTS

K.J.L. acknowledges baseline research funding from King Abdullah University of Science and Technology (KAUST). J.L.M. acknowledges DOE Office of

Science, Office of Biological and Environmental Research. External support for Macquarie University's Synthetic Biology initiative is acknowledged from Bioplatforms Australia, the New South Wales (NSW) Chief Scientist and Engineer, and the NSW Government's Department of Primary Industries. Australian Government funding through its investment agency, the Australian Research Council, toward the Macquarie University-led ARC Center of Excellence for Synthetic Biology is gratefully acknowledged by H.D.G. Figures 1, 3, and 4 were created by Heno Hwang, scientific illustrator at KAUST.

**DECLARATION OF INTERESTS**

J.L.M. is a paid consultant for Phycoil Biotechnology International, Inc., and a shareholder in Phycoil Biotechnology Korea, Inc. J.L.M. is an inventor on three patents related to *Prototheca moriformis* and *Auxenochlorella protothecoides* engineering that are listed in the references.

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
