## [Document S1. Transparent peer review records for Goold et al · Cell Genomics]

Cell Genomics, Volume 4

Supplemental information

The synthetic future of algal genomes

Hugh D. Goold, Jeffrey L. Moseley, and Kyle J. Lauersen

*THE SYNTHETIC FUTURE OF ALGAL GENOMES*Hugh D. Goold^{1,2,3}; Jeffrey L. Moseley^{4,5,6}; Kyle J. Lauersen⁷

Summary

Initial submission: Received : 9/28/2023
Scientific editor: Judith Nicholson

First round of review: Number of reviewers: 2
Revision invited : 12/6/2023
Revision received : 12/18/2023

Second round of review: Number of reviewers: N/A
Accepted : 1/24/2023

Data freely available: N/A

Code freely available: N/A

This transparent peer review record is not systematically proofread, type-set, or edited. Special characters, formatting, and equations may fail to render properly. Standard procedural text within the editor's letters has been deleted for the sake of brevity, but all official correspondence specific to the manuscript has been preserved.

Referees' reports, first round of review

Reviewer #1: This short review by Goold et al. presents a timely discussion of 'where we currently are' and 'where we might go in the near future' in the field of synthetic genomics as applied to eukaryotic microalgae. Overall, the manuscript is well presented, with a balanced and thoughtful overview of the current status of algal genomics, the potential of different microalgal genera/species as platforms, and how current whole genome re-engineering efforts in yeast and bacteria could be applied to such a platform in the future. I have only a few relatively minor comments for the authors:

1. [Abstract] "Advances in genome..." I would argue that improvements in DNA delivery technologies are also required.
2. [p3. 3.1/3.2] I have never seen the Glaucophyta described in the literature as 'blue algae'. In fact they are blue-green in hue as they also have chlorophyll a. However, calling them 'blue-green algae' would confuse the reader as this is the old name for the cyanobacteria. I would put blue in inverted commas in the 3.2 title, and just explain in the text that they are blue-green in colour.
3. [p3. 3.2] "unique...peptidoglycan walls". This is not strictly the case as there is evidence of remnant PG walls around the plastids of certain groups of land plants and some Chlorophyte algae (e.g. see: <https://link.springer.com/article/10.1007/s00709-023-01886-y>).
4. [p3 and elsewhere] Spelling of 'phosphorus'
5. [p3 bottom of para] '...or microorganism' Not sure what is meant here. Microscopic multicellular species ?
6. [p3 bottom of para] '...traditional crops' Again, unclear. Are you referring to land plants, and are you comparing PBRs or microalgae to these?
7. [p4. 3.2.1] '...adequate light'. Adequate CO₂ supply is also a challenge given the issues of efficient mass transfer.
8. [p4. 3rd para] I think you mean to refer to figure 3 ?
9. [p5. 4.1.1.] You refer to chloroplast transformation of *C. merolae* 10D twice.
10. [p5. 4.2] '...on acetic acid' Better to say 'on a medium containing acetate as a fixed carbon source'
11. [p5. 4.2] '...polyethylene glycol' It is the presence of abrasive glass beads rather than PEG that is key to transformation (PEG can be omitted and transformants still arise)
12. [p5. 4.2] '*C. reinhardtii* maintains separate..' So do virtually all other plastid-bearing eukaryotes: it's not unique to *C.r.* Maybe rephrase?
13. [p5. 4.2.1] 'The plastid genomes..are..' Use of plural implies that there are different genomes. Again, rephrase?
14. [p6, 1st para] 'the speed with which...' Not really correct to state that cp transformation is slower since nuclear transformation typically involves screening multiple lines because of messy integration, position effects and gene silencing. Also, methods for marker recycling in the cp are now available (e.g. <https://onlinelibrary.wiley.com/doi/10.1002/biot.202200088>)
15. [p8. 5.2.1] Do you mean Figure 3 here?
16. [p10. 5.2.4] And here?

Reviewer #2: Goold, Moseley and Lauersen have produced one of the most authoritative and comprehensive reviews of the last 2 years, dealing with advances in the field of algae biotechnology. Reading this review was a welcome change from the cacophony of opinions presented in various articles that rehashed the same

lines and produced little in terms of advancing the collective understanding or thinking.

The authors provided historical context for how the field has evolved, detailing key advances that produced paradigm shifts in engineering of algal genomes. This article reflects the current state of molecular technologies and basic understanding of different microalgae. The authors challenge the readers to use inspiration from initiatives like The Yeast 2.0 project, to push boundaries of what's currently deemed possible by many in the community.

I think the following minor corrections/suggestions will further improve this article:

- 1) Pg4, "from hydrogen and diesel biofuels (76), fragrances, flavours and cosmetics (81), and pharmaceuticals (145)." > Refs 76 and 145 are review and commentary respectively, therefore in the text the authors should indicate this as such (e.g. as reviewed in...) or cite the original research articles.
- 2) Pg4, "In an ideal scenario, an engineered alga would yield multiple products simultaneously" > But algae do produce multiple products...authors should improve this sentence to clarify their meaning, which I expect refers to engineered high-value products
- 3) Pg4, where Ref 29 and 157 are cited, and throughout the text > The authors could include other examples of research from the community. While the authors are respected authorities in the field, excessive self-citation will I fear lessen the broad reach of this review and that would indeed be unfortunate.
- 4) Pg4, "However, new species with favourable genetic and growth characteristics" > new species infers they have recently evolved, but I think the authors are alluding to prospected species that have recently come to light. Suggest change to "...knowledge of new species".
- 5) Pg5, "similar chloroplasts to plants have meant that *C. reinhardtii* has been adopted as a model for photosynthesis, starch biosynthesis, as well as circadian rhythm, microtubule formation and flagellar movement..." > sentence is too broad and should be broken up as similarity of chloroplast between plants and *Chlamydomonas* has not resulted in the latter serving as model for flagellar movement in plants!
- 6) Pg6, "completely recode its genome in several research groups (Saul Purton and Tobias Erb, personal communications)" > authors have indicated this impressive work to be a community effort (i.e. several research groups suggests more than the 2 listed), given the gravity of what these groups are setting out to do, I encourage authors to list as many of the groups involved as possible.
- 7) Pg7, "terpenoids such as isoprene (157) and polyamines such as cadaverine (30). These efforts illustrate the utility of engineered algae for producing specialty chemicals, particularly in the context of wastewater valorisation (29)." > again, these are important and ground-breaking work, but for the sake of broadening the reach of this review, perhaps it's worth citing more widely, e.g. Tran et al 2013 or Charoonnart et al 2019
- 8) Pg8, "All transformation and modifications goals" > "...modification..."
- 9) Pg9, "cultivation is straightforward" and Pg11 "doubling times are still long compared to rapid growers" > these statements, and those like these elsewhere in the text, when comparing growth parameters between species are non-descript and not helpful. To help a wider non-expert readership to benefit from this review, I suggest the inclusion of doubling time, cultivation conditions for all 7 species in a uniform fashion.
- 10) Pg10, "(Craig, R. et al, 2023, in preparation)" > as with all other references, replace with reference number, move text to References section and elaborate on co-authors and title of article.
- 11) Pg.12 "To date, only one study has demonstrated genetic manipulation in *Galdieria*, so continuing efforts are warranted" > Should a similar statement not be added for HR in *O. tauri*?

Authors' response to the first round of review

Reviewers' Comments:

Reviewer #1: This short review by Goold et al. presents a timely discussion of 'where we currently are' and 'where we might go in the near future' in the field of synthetic genomics as applied to eukaryotic microalgae. Overall, the manuscript is well presented, with a balanced and thoughtful overview of the current status of algal genomics, the potential of different microalgal genera/species as platforms, and how current whole genome re-engineering efforts in yeast and bacteria could be applied to such a platform in the future. I have only a few relatively minor comments for the authors:

Authors: Thank you for taking the time to review our manuscript in detail and your positive assessment of the work. We have updated the manuscript with each one of your corrections, in addition to other checks, which have improved the manuscript

substantially.

Reviewer #1: [Abstract] "Advances in genome..." I would argue that improvements in DNA delivery technologies are also required.

Authors: Thank you, we have incorporated this statement in the text.

Reviewer #1: 2. [p3. 3.1/3.2] I have never seen the Glaucophyta described in the literature as 'blue algae'. In fact they are blue-green in hue as they also have chlorophyll a. However, calling them 'blue-green algae' would confuse the reader as this is the old name for the cyanobacteria. I would put blue in inverted commas in the 3.2 title, and just explain in the text that they are blue-green in colour.

Authors: Thank you for this we have removed this incorrect statement and referred to this group only as Glaucophyta

Reviewer #1: [p3. 3.2] "unique...peptidoglycan walls". This is not strictly the case as there is evidence of remnant PG walls around the plastids of certain groups of land plants and some Chlorophyte algae, see:
<https://link.springer.com/article/10.1007/s00709-023-01886-y>

Authors: Thank you, we have included the suggested reference and updated the text to read: "The blue-green hued Glaucophyta plastids, called cyanelles, retain bacterial-like peptidoglycan walls⁷, which are found also in some chlorophytes, byrophytes and tracheophytes⁸."

Reviewer #1: 4. [p3 and elsewhere] Spelling of 'phosphorus'

Authors: Thank you for noting this, phosphorus is the noun, phosphorous is the adjective. We have harmonized as 'phosphorus' throughout the text.

Reviewer #1: 5. [p3 bottom of para] '...or microorganism' Not sure what is meant here. Microscopic multicellular species ?

Authors: Sentence updated to state "Algal biotechnology mainly focuses on unicellular or multicellular microbial species cultivated in photobioreactors of various dimensions or open pond systems¹¹."

Reviewer #1: 6. [p3 bottom of para] '...traditional crops' Again, unclear. Are you referring to land plants, and are you comparing PBRs or microalgae to these?

Authors: Sentence updated to state "Enclosed systems such as photobioreactors and fermenters offer superior control over culture conditions, invasive species and pests, but are more expensive than open (pond) systems¹³. Algae, in open or closed photobioreactors, can be cultivated on non-arable land using brackish or salt water and can have higher theoretical productivity in biomass per unit time than traditional land crops^{14,15}."

Reviewer #1: 7. [p4. 3.2.1] '...adequate light'. Adequate CO₂ supply is also a challenge given the issues of efficient mass transfer.

Authors: Thank you for noting this, we have updated the text to read: "Photosynthetic growth, while advantageous from a sustainability perspective, also necessitates infrastructure considerations to ensure adequate light penetration and carbon dioxide supply into cultivation vessels which generally yields lower volumetric cell densities than fermentation."

Reviewer #1: 8. [p4. 3rd para] I think you mean to refer to figure 3 ?

Authors: Thank you for noting this mistake, yes, and updated in the text.

Reviewer #1: 9. [p5. 4.1.1.] You refer to chloroplast transformation of *C. merolae* 10D

twice.

Authors: Thank you for noting this mistake, rectified.

Reviewer #1: 10. [p5. 4.2] '...on acetic acid' Better to say 'on a medium containing acetate as a fixed carbon source'

Authors: Thank you for noting this, we have updated the text to read: "... or heterotrophically on media containing acetate as a fixed carbon source, and is easy to isolate into axenic cultures."

Reviewer #1: 11. [p5. 4.2] '...polyethylene glycol' It is the presence of abrasive glass beads rather than PEG that is key to transformation (PEG can be omitted and transformants still arise)

Authors: The text has been modified to reflect this: "Transformation of DNA into *C. reinhardtii* can be achieved by agitation in the presence of glass bead or other particles, biolistic bombardment with DNA coated nanoparticles, and electroporation⁴⁵."

Reviewer #1: 12. [p5. 4.2] '*C. reinhardtii* maintains separate..' So do virtually all other plastid-bearing eukaryotes: it's not unique to *C.r.* Maybe rephrase?

Authors: Thank you for noting this, we have updated the text to read: "As is typical of primary endosymbiotic algae and plants, *C. reinhardtii* cells maintain separate nuclear and plastid genomes, each harbouring unique features and potentials for genetic engineering"

Reviewer #1: 13. [p5. 4.2.1] 'The plastid genomes..are..' Use of plural implies that there are different genomes. Again, rephrase?

Authors: Thank you for noting this, the plural was unintentional. We have updated the text to read: "The plastid genome of *C. reinhardtii* is evolutionarily linked to those of modern-day plants²."

Reviewer #1: 14. [p6, 1st para] 'the speed with which...' Not really correct to state that cp transformation is slower since nuclear transformation typically involves screening multiple lines because of messy integration, position effects and gene silencing. Also, methods for marker recycling in the cp are now available (e.g. <https://onlinelibrary.wiley.com/doi/10.1002/biot.202200088>)

Authors: Including the reference indicated, we have combined two sentences to read: "Numerous recombinant proteins have been produced in this organelle and iterative transformations have been made possible development of chloroplast marker recycling methods⁵⁸."

Reviewer #1: 15. [p8. 5.2.1] Do you mean Figure 3 here?

Authors: Thank you for noting this mistake, yes, and rectified.

Reviewer #1: 16. [p10. 5.2.4] And here?

Authors: Thank you for noting this mistake, yes, and rectified.

Reviewer #2: Goold, Moseley and Lauersen have produced one of the most authoritative and comprehensive reviews of the last 2 years, dealing with advances in the field of algae biotechnology. Reading this review was a welcome change from the cacophony of opinions presented in various articles that rehashed the same lines and produced little in terms of advancing the collective understanding or thinking. The authors provided historical context for how the field has evolved, detailing key advances that produced paradigm shifts in engineering of algal genomes. This article

reflects the current state of molecular technologies and basic understanding of different microalgae. The authors challenge the readers to use inspiration from initiatives like The Yeast 2.0 project, to push boundaries of what's currently deemed possible by many in the community.

Authors: Thank you for the kind assessment of our work. We appreciate the recognition of a forward-looking vision of our field.

Reviewer #2: I think the following minor corrections/suggestions will further improve this article:

1) Pg4, "from hydrogen and diesel biofuels (76), fragrances, flavours and cosmetics (81), and pharmaceuticals (145)." > Refs 76 and 145 are review and commentary respectively, therefore in the text the authors should indicate this as such (e.g. as reviewed in...) or cite the original research articles.

Authors: Thank you for noting this, we have updated the text here to cite specifically reviews on the topics, which now reads:

"Nevertheless, these challenges present opportunities, and several research groups have modified diverse species of algae to make various products, covered in multiple reviews: from hydrogen and diesel biofuels²⁰, fragrances, flavours and cosmetics^{18,21}, and pharmaceuticals^{22,23}."

Reviewer #2: 2) Pg4, "In an ideal scenario, an engineered alga would yield multiple products simultaneously" > But algae do produce multiple products...authors should improve this sentence to clarify their meaning, which I expect refers to engineered high-value products

Authors: Thank you for noting this, we intended to indicate the value-add of a strain engineered to make multiple recombinant products and/or other native compounds. We have clarified this in the text as follows:

"Native natural products can be sourced from algal biomass or strain genetic engineering can be used to enhance yields of these products or generate non-native biochemicals in the algae. Algal biomass can be separated into hydrophobic and hydrophilic chemicals, in addition to possible secreted products such as extracellular polysaccharides. Sourcing multiple products of value from a single cultivated organism is promoted as a concept termed 'bio-refinery'. This approach is traditionally discussed while using algae to clean wastewaters of inorganic contaminants while simultaneously generating multiple products from the algal biomass. Engineered algae applied in this context could yield multiple targeted products simultaneously, both native and engineered. For example, ..."

Reviewer #2: 3) Pg4, where Ref 29 and 157 are cited, and throughout the text > The authors could include other examples of research from the community. While the authors are respected authorities in the field, excessive self-citation will I fear lessen the broad reach of this review and that would indeed be unfortunate.

Authors: Thank you for noting this, as our original submission requirements for this manuscript limited total citations, we were not as broad as we could have been with referencing the available literature. We have made an active effort to expand the referenced literature for engineered bio-products made in *Chlamydomonas* by other groups.

Reviewer #2: 4) Pg4, "However, new species with favourable genetic and growth characteristics" > new species infers they have recently evolved, but I think the authors are alluding to prospected species that have recently come to light. Suggest change to "...knowledge of new species".

Authors: Thank you for noting this, we have updated the text to read: "However, improving knowledge of newly characterized species with favourable genetic and growth characteristics is expanding the repertoire of algae that are suitable for engineering approaches."

Reviewer #2: 5) Pg5, "similar chloroplasts to plants have meant that *C. reinhardtii* has

been adopted as a model for photosynthesis, starch biosynthesis, as well as circadian rhythm, microtubule formation and flagellar movement..." > sentence is too broad and should be broken up as similarity of chloroplast between plants and Chlamydomonas has not resulted in the latter serving as model for flagellar movement in plants!

Authors: Thank you for noting this, we have updated the text to read: "Its similar chloroplasts to plants have meant that *C. reinhardtii* has been adopted as a model for photosynthesis, and beyond that it has been used to study starch biosynthesis, circadian rhythms, microtubule formation and flagellar movement, sexual reproduction, and the light-gated ion channels discovered in *Chlamydomonas* opened up the field of optogenetics⁴⁴."

Reviewer #2: 6) Pg6, "completely recode its genome in several research groups (Saul Purton and Tobias Erb, personal communications)" > authors have indicated this impressive work to be a community effort (i.e. several research groups suggests more than the 2 listed), given the gravity of what these groups are setting out to do, I encourage authors to list as many of the groups involved as possible.

Authors: Thank you for noting this, we have expanded the section to highlight efforts by the groups of Saul Purton - University College London, Alison Smith and Jason Chin - Cambridge⁵⁹, Tobias Erb - Max-Planck-Institute for Terrestrial Microbiology - Marburg, Duanmu Deqiang - Huazhong Agricultural University, Wuhan, and Zhangli Hu, Chaogang Wang, and Bin Jia - Shenzhen University⁶⁰

Reviewer #2: 7) Pg7, "terpenoids such as isoprene (157) and polyamines such as cadaverine (30). These efforts illustrate the utility of engineered algae for producing specialty chemicals, particularly in the context of wastewater valorisation (29)." > again, these are important and ground-breaking work, but for the sake of broadening the reach of this review, perhaps it's worth citing more widely, e.g. Tran et al 2013 or Charoonnart et al 2019

Authors: Thank you for noting this, we have made an active effort to expand the referenced literature for engineered bio-products made in *Chlamydomonas* by other groups. This sentence now is followed by "Biotherapeutic applications include engineered biocontrol of pathogens in aquaculture⁹², and expression of anti-cancer immunotoxins⁵⁰."

Reviewer #2: 8) Pg8, "All transformation and modifications goals" > "...modification..."

Authors: Thank you for noting this, we have updated the text.

Reviewer #2: 9) Pg9, "cultivation is straightforward" and Pg11 "doubling times are still long compared to rapid growers" > these statements, and those like these elsewhere in the text, when comparing growth parameters between species are non-descript and not helpful. To help a wider non-expert readership to benefit from this review, I suggest the inclusion of doubling time, cultivation conditions for all 7 species in a uniform fashion.

Authors: We have added specific doubling times and, where possible, growth modes of each organism with references in each section.

Reviewer #2: 10) Pg10, "(Craig, R. et al, 2023, in preparation)" > as with all other references, replace with reference number, move text to References section and elaborate on co-authors and title of article.

Authors: completed as requested: Ref 148.

Reviewer #2: 11) Pg.12 "To date, only one study has demonstrated genetic manipulation in *Galdieria*, so continuing efforts are warranted" > Should a similar statement not be added for HR in *O. tauri*?

Authors: text updated to read: "Efficient HR, requiring short homology arms for gene

targeting, has been reported only once¹¹⁸, and may require further characterisation.”